# State-of-the-Art Differentiation Protocols for Patient-Derived Cardiac Pacemaker Cells

**DOI:** 10.3390/ijms25063387

**Published:** 2024-03-16

**Authors:** Eleonora Torre, Matteo E. Mangoni, Alain Lacampagne, Albano C. Meli, Pietro Mesirca

**Affiliations:** 1Institut de Génomique Fonctionnelle, Université de Montpellier, CNRS, INSERM, 34090 Montpellier, France; eleonora.torre@igf.cnrs.fr (E.T.); matteo.mangoni@igf.cnrs.fr (M.E.M.); 2LabEx Ion Channels Science and Therapeutics (ICST), 06560 Valbonne, France; 3PhyMedExp, University of Montpellier, Inserm, CNRS, 371 Avenue du Doyen G. Giraud, CEDEX 5, 34295 Montpellier, France; alain.lacampagne@inserm.fr (A.L.); albano.meli@inserm.fr (A.C.M.); 4Montpellier Organoid Platform, Biocampus, University of Montpellier, CNRS, INSERM, 34090 Montpellier, France

**Keywords:** sinoatrial node, hiPSC-derived cardiac pacemaker cells, protocols

## Abstract

Human-induced pluripotent stem cell (hiPSC)-derived cardiomyocytes raise the possibility of generating pluripotent stem cells from a wide range of human diseases. In the cardiology field, hiPSCs have been used to address the mechanistic bases of primary arrhythmias and in investigations of drug safety. These studies have been focused primarily on atrial and ventricular pathologies. Consequently, many hiPSC-based cardiac differentiation protocols have been developed to differentiate between atrial- or ventricular-like cardiomyocytes. Few protocols have successfully proposed ways to obtain hiPSC-derived cardiac pacemaker cells, despite the very limited availability of human tissues from the sinoatrial node. Providing an in vitro source of pacemaker-like cells would be of paramount importance in terms of furthering our understanding of the mechanisms underlying sinoatrial node pathophysiology and testing innovative clinical strategies against sinoatrial node dysfunction (i.e., biological pacemakers and genetic- and pharmacological- based therapy). Here, we summarize and detail the currently available protocols used to obtain patient-derived pacemaker-like cells.

## 1. Introduction

Heart automaticity is a fundamental physiological function that is reliant on the presence of a highly specialized population of cardiomyocytes in the sinoatrial node (SAN). These cells are referred to as pacemaker cells [1]. The spontaneous activity of pacemaker cells is due to diastolic depolarization (DD), a slow depolarization phase that drives the membrane potential from the end of an action potential (AP) to the threshold of a new AP. SAN cells express a wide array of ion channels, which underlie the generation and regulation of DD by the autonomic nervous system (ANS) [1]. Knowledge of the functional role of ion channels in pacemaker activity mostly comes from small-animal models (i.e., rodents) [2,3,4]. Human biospecimens cannot fully replace animal models in cardiovascular research because of the difficulty of performing the genetic manipulation of native human myocytes. However, associations between data obtained in rodent models and insights obtained from human-derived cells and cardiac tissue are important in terms of improving the translational significance of preclinical research [5]. The availability of native human cardiac tissue samples may be limited, especially tissue from individuals with congenital heart diseases that do not require cardiac reparative surgery. Indeed, researchers commonly obtain samples during surgical septal myectomy and cardiac catheterization from heart transplants. However, the collection of SAN biopsies from patients for research purposes is currently impossible because of the risk of irreversibly damaging automatic tissue, a concern associated with surgical procedures. Consequently, living human SAN tissue samples are not widely available and are mostly obtained from cardiac graft recipients’ diseased hearts. Searching for complementary sources of human-derived cardiomyocytes is thus important. In this context, the differentiation of human-induced pluripotent stem cells (hiPSCs) into cardiomyocytes (hiPSC-CMs) contributes to lessening the limitations in the availability of in vitro human models and may constitute a paramount breakthrough in the fields of disease modeling and personalized therapy [6,7]. Despite hiPSCs being a different source of human cells than embryonic stem cells, some ethical limitations are to be considered. First, the collection and harvesting of source cells in order to produce hiPSCs requires patients’ written consent and ethical approval. This procedure implies the confidential storage of information about the history of the patient’s disease and their personal history. Another ethically challenging aspect is related to the usage of hiPSCs for basic research and/or therapy. Indeed, most hiPSC-CM research is devoted to modeling the idiopathic or inherited pathology of heart rhythm. This approach does not provide immediate or even short-term benefits to the patient. Consequently, when providing information to the patient to obtain written consent, the researcher needs to clearly state that research using the patient’s cellular source will not affect or improve therapeutic approaches to the pathology. This aspect brings substantial communication challenges especially, when the given pathology may have a fatal outcome. In addition, access to patients’ cellular sources may be limited by ethics in the case of secondary and acquired arrhythmias and when studying potential risk factors because storing patients’ history information that is not directly related to their primary pathology may be considered a violation of privacy. Nevertheless, the study of several hiPSC lines derived from patients has considerably advanced our knowledge of the mechanisms of primary arrhythmias. While in vitro models of ventricular arrhythmias have been published [8,9], no hiPSC-based model of primary sinoatrial node dysfunction is currently available. In addition, while several efficient protocols exist for differentiating ventricular-like hiPSC-CMs [9,10], protocols for differentiating pacemaker sinoatrial node-like cells (hiPSC-PMs) remain poorly developed.

The purpose of this review article is to discuss the developmental and molecular basis of the published protocols used to differentiate hiPSC-PMs. In this context, we summarize the transcriptional cascade underlying SAN biogenesis in order to highlight the key molecules that are important in promoting specification towards cardiac pacemaker cells. We also discuss important electrophysiological hallmarks of the differentiation of hiPSCs into hiPSC-PMs. Finally, we indicate potential research avenues via which to improve differentiation into the hiPSC-PM phenotype.

## 2. Genesis of SAN: Molecular Determinants

### 2.1. Development of Cardiac Mesoderm

During differentiation, hiPSCs are treated using different molecules to promote cardiac specification. Defining molecules involved in hiPSC differentiation is strictly dependent on knowledge of the genesis of the heart.

In embryo development, gastrulation is a key event that begins with the generation of a transient structure called a primitive streak, through which the three germ layers (endoderm, mesoderm, and ectoderm) are formed [11]. The primitive streak results from the proliferation and the migration of the epiblast cells into the median plane of the embryonic disc. As the cells invaginate, some of them displace the underlying hypoblast, forming the embryonic endoderm. Some of the invaginated cells of the primitive streak are found between the epiblast and newly formed endoderm, generating the intra-embryonic mesoderm. Later, the remaining cells of the epiblast give rise to the ectoderm (Figure 1) [12,13,14,15]. The mammalian heart derives the lateral plate mesoderm from a specific region of mesoderm.

The formation of the mesoderm and its transition to a cardiac mesoderm occur due to Wnt/β-catenin signaling [16]. Wnt/β-catenin signaling is activated when Wnt binds to its receptor, which induces β-catenin stabilization. β-catenin migrates into the nucleus in order to regulate target genes through its interaction with the TCF/LEF transcription factors. In the absence of Wnt signaling or the presence of an inhibitor such as the endoderm of Dickkopf Wnt Signaling Pathway Inhibitor 1 (DKK1), β-catenin is degraded by a protein complex, preventing target gene regulation.

In particular, mesoderm specification is induced by the activation of Wnt/β-catenin signaling (Figure 1). At post coitum embryonic (E) day 5.75 (E5.75) in developing mice (corresponding roughly to week 2 of human gestation), activated Wnt/β-catenin signaling induces the expression of mesendodermal markers, such as Brachyury (Bry) [17] and Eomesodermin (EOMES) [18,19]. These directly activate the primary cardiac mesoderm regulator, mesoderm posterior BHLH transcription factor 1 (Mesp1) [20,21], and the family member Mesp2 [22]. This mechanism is reinforced by the activation of TGF-β superfamily signaling by Nodal and BMP4 (bone morphogenetic protein 4) agonists [23,24]. Together with Mesp1/2, Nodal and BMP4 induce the expression of the platelet-derived growth factor receptor-α (Pdgfr-α), which can be used to effectively monitor the emergence of cardiovascular mesoderm [25]. In the anterior part of the mesoderm, Mesp1/2 activates the secretion of DKK1 from the endoderm (Figure 1). DKK1, which subsequently inhibits Wnt/β-catenin signaling, converts the anterior mesoderm into a cardiogenic mesoderm (Figure 1) [20,21,26]. Together with Wnt/β-catenin signaling inhibition, BMP4 and the fibroblast growth factor (FGF) support cardiac mesoderm formation [23,27]. In the posterior component, where Wnt/β-catenin signaling is still active and notochord (axial mesoderm)-released Noggin inhibits BMP4, the lateral plate mesoderm becomes the hemangiogenic mesoderm [28,29].

### 2.2. First and Second Heart Fields

In mouse E8, corresponding to 3 weeks of human gestation, the specification of the cardiac lineage begins when cardiac progenitor cells (CPCs) migrate anterolaterally from cardiogenic mesoderm and form two different fields, known as the first and second heart fields (FHF and SHF, respectively). At this point, cardiac transcription factors became fundamental markers used to define the different populations.

The first pool of migrant cells becomes the FHF lineage: FHF cells are identified by the expression of Nkx2-5 (NK2 transcription factor related, locus 5 [Drosophila]), Tbx5 (T-box transcription factor 5), HCN4 (hyperpolarization-activated cyclic nucleotide-gated potassium channel 4), and Hand1 (Heart- and neural crest derivatives-expressed protein 1), among others markers [30]. FHF cells have been identified as contributing to the myocardium of the left ventricle (LV), atrioventricular canal, and the proepicardium. Together with cardiac mesoderm specification, FGF is also required in the maintenance of chamber identity in developing ventricles, suppressing cardiomyocyte plasticity at this stage and even in differentiated cardiomyocyte fields [31,32].

Later, SHF development depends on the use of instructive Wnts sourced from the already formed FHF [33]. In particular, the Wnt/β-catenin signaling regulates the expression of the LIM (Lin11, Isl1, and Mec3) homeodomain transcription factor Islet1 (Isl1) in the SHF. The SHF segregates into an anterior component (aSHF) and a posterior component (pSHF) with different fates. aSHF cells give rise to the right ventricle (RV) and the outflow tract (OFT), while cells of pSHF contribute to the formation of the atria first, followed by the inflow tract (IFT) region or venous pole. From the IFT, the sinus horns and the primordial SAN are formed. At day E10.5 in mice (day 32 in humans), the developing heart shows well-defined chambers (Figure 1).

### 2.3. SAN Generates from Posterior SHF

SAN development is initiated in the presence of Isl1-expressing SHF-derived cells. However, the imposition of the posterior limit of the SHF is, in part, controlled by retinoic acid (RA) signaling, which restricts the border of aSHF while maintaining its posterior component (Figure 1) [34]. Knockout mice deficient in retinaldehyde dehydrogenase 2 (Raldh2), which catalyzes the second oxidative step in RA biosynthesis, show alteration in terms of SHF orientation [35]. aSFH expresses Hox genes as well as Tbx1, and these antagonize RA signaling in this region. In the pSHF, RA is essential to restricting the expression of Nkx2-5. This is suggested by the observation that RA deficiency leads to an enlargement of the early domains expressing Nkx2-5 [35].

Moreover, pSHF itself is partitioned into right and left pSHF. In particular, Pitx2c expression in the left pSHF contributes to atria development [36,37]. In the right side of the pSHF, where Pitx2c is downregulated by Nodal [38], a negative regulation of Nkx2-5 is established, leading to the expression of Tbx3. Consequently, in the right pSHF, Tbx3 inhibits chamber myocardial differentiation, preventing the expression of Nppa (natriuretic peptide precursor type A) and Cx40 (Connexin 40) proteins, which are specific atrial chamber myocardial markers. Tbx3 not only represses genes required to differentiate the working myocardium and limit the expression of atrial chamber markers, but also stimulates genes required for pacemaker function, including HCN4, L-type Ca^2+^ channel, and slow gap junction Cx30.2, efficiently reprogramming atrial myocytes into functional pacemaker cells [37,39]. Notably, the expression of HCN4 in FHF initially overlaps with that of Nkx2-5, but subsequently becomes restricted to the newly recruited Nkx2-5-negative venous pole components [40]. The population of cells co-expressing Tbx18, Tbx3, and HCN4, but not expressing Cx40 and Nppa, constitutes the precursor population to SAN specification [41]. The three-dimensional reconstruction of expression patterns show that, during heart tube elongation, the Tbx18^+^ progenitors remain spatially and temporally separate from the Isl1^+^ SHF. They only overlaps with the Isl1^+^ domain at the right lateral side of the IFT, where the SAN develops (Figure 1) [42]. Isl1 is regulated in a synergic way by Tbx5 via Shox2 (short-stature homeobox protein 2) present in the IFT [43,44]. Biochemistry experiments also demonstrate also that the Isl1^+^ population requires Shox2 to repress the Nkx2-5 [45,46] and positively regulates BMP4 for the subsequent SAN formation [43,47]. BMP4 seems to upregulate Tbx18 expression via Gata4 and supports the downregulation of Nkx2-5 [48].

### 2.4. Map of SAN Markers

Defining critical markers involved in heart development and the differentiation of the SAN could help to orientate the differentiation of hiPSCs into hiPSC-PMs.

While the SAN is generally considered to be a unique region of the heart, structural studies have distinguished three different SAN sub-regions: the head, center, and tail [49]. These different subregions correspond in the adult SAN to a different spectrum of markers that defines these territories. The transcriptional landscape of mouse SANs was revealed via single-cell RNA sequencing (scRNA-seq) [50]. The SAN head is defined using the principal markers that we listed in Section 2.3: Isl1^+^, Shox2^+^, Tbx3^+^, Tbx18^+^, Nppa^−^, Nkx2-5^−^, and Cx40^−^. Few differences characterized the tail region (i.e., Tbx18^−^, Nkx2-5^+^) [50]. No information is available on the central part, which is supposed to be characterized by a mixed spectrum of markers from the central and the tail regions.

## 3. Electrical Activity of the Native SAN

Along with markers of cardiac development, adult SAN is characterized by a specific pattern of ion channels that ensures the generation of spontaneous activity (Figure 2). The generation of automaticity in cardiac pacemaker cells is due to DD, a spontaneous, slowly depolarizing phase of the AP cycle [1]. During this phase, the membrane potential progressively becomes less negative until it reaches the threshold for triggering a new AP. One of the key initiators of the DD is the hyperpolarization-activated “funny current” (I_f_), an inward Na^+^/K^+^ current predominantly carried by the HCN4 isoform in rodents [3]. In humans, the expression of HCN isoform 1 (HCN1) is higher in the SAN than in HCN4 [51]. Consequently, HCN1 could be useful as a new specific molecular marker of the human SAN.

Moreover, SAN myocytes constitute a unique type of cardiac cell as they co-express two functionally distinct isoforms of L-type Ca^2+^ channels: the cardiac Ca_v_1.2 (Cacna1C) isoform, which couples excitation to contraction in the working myocardium [52], and the Ca_v_1.3 (Cacna1D) isoform, which contributes to SAN DD [2]. Together with L-type Ca^2+^ channels, the T-type Ca^2+^ channel Ca_v_3.1 contributes to DD [53]. The AP upstroke of pacemaker cells is mainly driven by Ca^2+^ rather than Na^+^ channels (I_Na_) [54]. In the rabbit SAN, I_Na_ does not participate in the generation of automaticity per se (in the central SAN), but it can influence heart rate (HR) by contributing to impulse propagation in the SAN and from the SAN to the atrium [55]. The activation of Ca_v_1.3 triggers sarcoplasmic reticulum (SR) Ca^2+^ release via ryanodine receptor 2 (RyR2) [56]. Ca^2+^ released from SR stimulates the activity of the Na^+^/Ca^2+^ exchanger (NCX1). This permits the restoration of diastolic Ca^2+^ content, generating an inward Na^+^ current.

The ANS has a decisive effect on HR acceleration and deceleration. In the adult heart, the sympathetic branch of the ANS accelerates HR, while the parasympathetic branch slows it down. SAN is enriched in adrenergic and muscarinic receptors [1]. The sympathetic regulation of pacemaker activity is mediated by catecholamines (i.e., norepinephrine), activating the β-adrenergic receptors (β_1/2_-ARs). The activation of β-ARs stimulates adenylyl cyclase (AC) activity, which converts ATP into cyclic AMP (cAMP). Elevated cAMP directly stimulates I_f_ via the cyclic nucleotide-binding domain (CNBD) of the HCN protein and activates the protein kinase A (PKA). The catalytic subunit of PKA enhances the activity of Ca_v_1.3 and Ca_v_1.2 channels via PKA-dependent phosphorylation of the cardiac L-type channel regulatory partner protein Rad [57,58,59]. The regulation of T-type Ca^2+^ channels by PKA is still controversial; however, it is shown that Ca_v_3.1 channels are stimulated by β-adrenergic agonists, probably via machinery similar to that used for L-Type Ca^2+^ channels [53]. The deceleration of HR is mediated by parasympathetic regulation of cardiac automaticity via the activation of muscarinic receptors (M2Rs) following the release of acetylcholine (ACh) from vagal nerve endings [1]. M2Rs are negatively coupled with AC activity. Thus, the downregulation of cAMP and, consequently, reductions in PKA phosphorylation reverse the signaling processes involved in the sympathetic stimulation of HR, during which I_f_, Ca_v_1.3/Ca_v_1.2, and Ca_v_3.1 are positively stimulated. In addition, M2Rs directly activate the ACh-activated outward K^+^ current (I_KACh_), which induces the hyperpolarization of the membrane potential and consequently reduces the pacemaker rate [60]. Two subunits of G-protein-activated inwardly rectifying K^+^ channels (GIRK1 and GIRK4) assemble as heterotetramers to form cardiac I_KACh_ channels [61].

The array of ion channels characterizing pacemaker cells can be considered valuable instruments in efforts to define the population of hiPSC-PMs. Biophysical properties that have been studied via single-cell recordings using patch clamp or planar lipid techniques are still informative and accurate tools, disclosing the mechanisms underlying electrical activity in hiPSC-PMs.

## 4. Protocols to Differentiate hiPSC-PMs

We used critical checkpoints in heart development to establish several protocols for the in vitro differentiation of hiPSC-PMs. Previously listed markers are used to validate the quality of the hiPSC-PM population. Electrophysiological properties are tested later to evaluate the phenotypes of these cells.

After elucidating the complexity of SAN development in terms of signaling and molecular markers, it became clear that the manipulation of single transcriptional factors alone could not be sufficient to induce the in vitro differentiation of hiPSC-PMs. Using this approach, two protocols were proposed by increasing the expression of Tbx3 [62] or Shox2 [63]. However, Tbx3 and Shox2 remain two downstream factors in heart development. Upstream events (i.e., cardiac mesoderm induction, FHF and SHF segregation) are important commitment steps to generate, as much as possible, a significant population of hiPSC-PMs in mixed cultures. In particular, it was the study by Zhu et al. [64] that first considered the regulation of an appropriate signaling pathway as an approach to enriching human embryonic stem cell-derived cardiomyocytes (hESC-CMs). Neuregulin (NRG)-1β, along with its receptor tyrosine kinase, ErbB, is involved in cardiac specification in the differentiation of hESC-CMs. Their role suggests that manipulating NRG-1β/ErbB signaling affects the ratio of Nodal- to working-type cells in differentiating hESC-CM cultures. However, reducing hESC-CM heterogeneity via the manipulation of this pathway did not consider several other pathways involved in SAN specification (i.e., Wnt/β-catenin and RA signaling), which are fundamental to imposing the presence or absence of specific tissue markers.

### 4.1. Transgene-Dependent Methods to Obtain hiPSC-PMs

Later, several groups tried to obtain hiPSC-PMs, combining the manipulation of pathways involved in heart development with cell sorting experiments to purify the population of interest (Figure 3).

The first strategy was proposed by the research group led by Christine Mummery [65]. These authors recognized the importance of precisely modulating the transition from CPCs to differentiated cardiomyocytes in improving cardiac lineage specification. In particular, they took advantage of the increasing expression of Pdgfr-α in CPCs to follow cardiac progression and the progression of podoplanin (PDPN), whose expression is largely restricted to the SAN [66]. In addition, they used hESCs, in which Nkx2-5 was associated with eGFP (Nkx2-5-eGFP), together with a doxycycline (dox)-inducible MYC transgene. They tried to restrain Nkx2-5 enhancement in hESCs, activating MYC transgene by dox at day 4.75 to obtain SAN-like cells. In order to reinforce Nkx2-5 downregulation, the Nodal pathway was inhibited (Figure 3). The authors selected a Nkx2-5-eGFP^−^ Pdgfr-α^+^ PDPN^+^ population. However, when maintaining Nkx2-5-eGFP^−^ Pdgfr-α^+^ PDPN^+^ cells in the self-renewal stage, the use of dox combined with Nodal inhibition was not sufficient. To amplify the Nkx2-5-eGFP^−^ Pdgfr-α^+^ PDPN^+^ population, mitogenic activators were added (i.e., insulin-like growth factor-1, IGF-1; hedgehog agonist, Hh). On day 12, MYC transgene was inactivated by removing dox in order to put Nkx2-5-eGFP^−^ Pdgfr-α^+^ PDPN^+^ cells through cardiac differentiation. During the cardiac differentiation stage, IGF-1 and Hh stimulation were maintained, as was Nodal pathway inhibition; moreover, the Wnt pathway was inhibited. In this way, Nkx2-5-eGFP^−^ Tbx3^+^ Shox2^+^ HCN4^+^ cells were obtained (Figure 3). Although this strategy produces a large number of cells, the overexpression of MYC may generate pro-oncogenic cell types that cannot be used in vivo to generate biological pacemakers.

Nkx2-5 was also used to monitor hiPSC differentiation in a protocol proposed by Gordon Keller’s group [67]. They started with two different cell lines: the HES3Nkx2-5^GFP/w^ cell line, similar to the Nkx2-5-eGFP hESC line, and a non-genetically modified hiPSC line. A previously described stepwise development protocol [25] was used to obtain a mixed population of hiPSC-CMs (Figure 3). Briefly, Bry^+^ hiPSCs were selected and cultured as embryonic bodies (EBs). EBs were maintained in culture in a low-oxygen environment until day 12. During this period, cells were first treated with FGF, BMP4, and Nodal to obtain Pdgfr-α^+^ Mesp1^+^ EBs. On day 3, Nodal and FGF pathways were inhibited while the BMP4 route remained active and supported by RA stimulation. On day 20, cells were sorted using Nkx2-5, SIRPA (pan-cardiac myocyte marker), and CD90 (mesenchymal marker). The Nkx2-5^−^ SIRPA^+^ CD90^−^ population expressed considerably higher levels of genes associated with SAN compared to the Nkx2-5^+^ population, including Isl1, Tbx3, Tbx18, Shox2, and HCN isoforms 1 and 4. The advantage of this protocol comes from the possibility of generating hiPSC-PMs for the development of clinically compliant biological pacemakers without the need for the genetic manipulation of hiPSCs.

**Figure 3 ijms-25-03387-f003:**
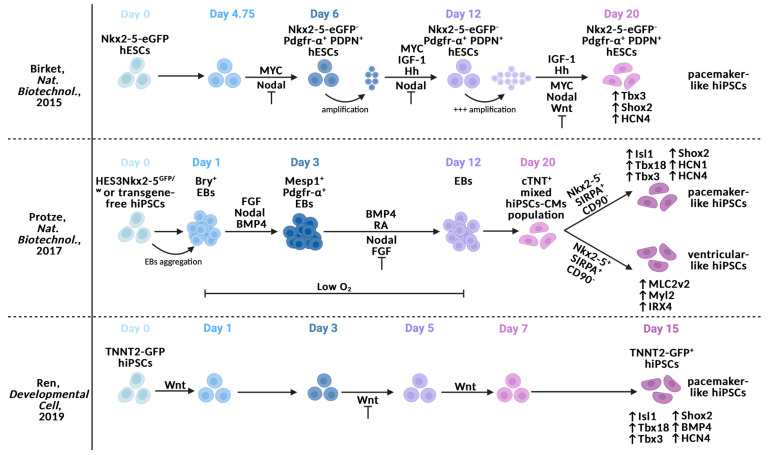
Summary of experimental designs used to induce hiPSC-PMs through transgene-based methods combined with cardiac development-based pharmacological manipulation [65,67,68].

Finally, Ren et al. [68] designed a pacemaker-like differentiation protocol based on the modulation of the Wnt pathway (Figure 3). Indeed, they showed that the re-activation of Wnt/β-catenin signaling was sufficient to enforce the cellular identity of the pacemaker phenotype. They validated this finding in a TNTT2-GFP hiPSC line. To this end, they induced mesoderm formation via the activation of the Wnt/β-catenin signaling from day 0 to day 1. The Wnt/β-catenin signal was inhibited from day 3 until day 5 to force cardiac mesoderm generation. Because they observed that the Wnt5b isoform was able to silence Nkx2-5 and upregulate SAN transcription factors (i.e., Isl1 and Tbx18) after cardiac mesoderm induction in zebrafish, they re-activated the Wnt/β-catenin signaling from day 5 until day 7. On day 15, the TNTT2-GFP^+^ cell population exhibited an increase in SAN markers (i.e., Isl1, Tbx18, Tbx3, Shox2, BMP4, and HCN4) and a reduction in Nkx2-5 expression with the re-activation of Wnt/β-catenin signaling compared to TNTT2-GFP^+^ cells population, where Wnt/β-catenin signaling was not re-activated.

### 4.2. Transgene-Free Methods to Obtain hiPSC-PMs

While these pioneering strategies were fundamental to providing a rich source of hiPSC-PMs, the high complexity of protocols that are reliant on genetic manipulation may reduce the chances of reproducibility, driving several groups to propose differentiation protocols based mainly on the pharmacological modulation of cardiac development pathways in order to obtain hiPSC-PMs (Figure 4).

Liu et al. [69] started with a cardiac differentiation protocol similar to the one of Ren et al. [68] (Figure 4): between day 0 and day 1, Wnt/β-catenin signaling was activated, being switched off from day 3 until day 5. On day 5, hiPSCs were treated with a BMP4 activator, FGF inhibitor, and a potent RArβ (RA receptor β) agonist that acts as an antagonist against RArα and RArγ. In particular, RArs are expressed widely [70,71]: RArα shows ubiquitous expression in the developing heart [72]. RArβ expression is restricted in the OFT mesenchyme and is found through the developing myocardium [72,73]. RArγ transcripts are specifically detected in the endocardial cushion tissue and large developing vessels [70]. Moreover, RArα activation seems to improve the atrial differentiation of hiPSCs [74]. In this way, FGF signaling, which is known to sustain ventricular development, was inhibited. BMP4 was added to support SAN differentiation, and RArβ-selective activation prevented atrial, endocardial, and vessel specification, promoting SAN differentiation. Cells treated with this method showed higher levels of expression of Shox2, Tbx3, Tbx18, HCN4, and TNNT2 in comparison to untreated ones.

A simplified version of this protocol was proposed by Yechikov [75] (Figure 4). After the activation of Wnt/β-catenin signaling between days 0 and 1, inhibition of Wnt/β-catenin signaling was accompanied by Nodal signaling inactivation between days 3 and 5. Yechikov based this protocol on the concept that Nodal is an upstream effector of Pitx2c and that the inhibition of Pitx2c via Nodal inactivation could promote hiPSC-PM differentiation. In addition to the simplicity of this protocol, treated cells exhibited increased transcript and protein expression of Tbx3, Tbx18, Shox2, and Isl1.

Wiesinger et al. [76] used a modified Protze-based protocol [67] to generate a roadmap of transcriptional changes during hiPSC-PM differentiation. Briefly, hiPSCs were initially directed towards a mesodermal lineage by supplementing the medium with BMP4, Wnt and Nodal activators; then hiPSCs were further guided towards a cardiac fate by inhibiting the Wnt/β-catenin signaling pathway (Figure 4). At day 4, alongside inhibiting Wnt/β-catenin signaling, the addition of BMP4 and RA in combination with Nodal and FGF inhibition, drove cardiac mesoderm to cardiomyocytes with a pacemaker-like profile. The obtained hiPSC-PMs exhibited higher expression levels of Shox2, Tbx3, Tbx18, HCN4, HCN1, Cacna1d, Cacna1g, and TNNT2 compared to the ventricular-like counterpart.

An exception to these methods was the protocol proposed by Schweizer [77] (Figure 4). They provided a transgene-free approach by co-culturing hiPSCs with mouse visceral endoderm-like (END-2) cells till day 12 in serum-free medium. After, beating clusters were dissected and transferred to dishes with serum-enriched medium. They observed a decrease in Tbx5 and Nkx2-5 expression with an increase in Shox2 expression. However, co-culturing hiPSC-PMs with END-2 cells has a high xenogeneic risk if we consider the possible medical applications. Moreover, interaction mechanisms between END-2 cells and hiPSCs are still unknown and it is difficult to predict how changes in END-2 cell culture can influence the hiPSCs differentiation. For these reasons, recently Schweizer group established four END2-cell independent protocols to obtain hiPSC-PMs [78]. Here, three protocols proposed combine Wnt/β-catenin signaling activation/inactivation, Nodal inactivation, and RA administration at different time points or combined in different ways (Figure 4). The fourth protocol (STEMCELL protocol) consists of a STEMdiff™ Atrial Cardiomyocyte Differentiation Kit from STEMCELL Technologies (STEMCELL protocol). Protocols are compared to the END-2 cells dependent one. SAN marker expression was higher using the STEMCELL protocol.

More recent findings bring the role of cadherin-5 protein (CDH5 or VE-Cadherin) on the differentiation of hiPSC-PMs [79]. CHD5 is a cell adhesion glycoprotein expressed in vascular endothelial cells [80]. Treatment with CDH5 during cardiac differentiation led to an increased proportion of hiPSC-PMs [79] (Figure 4). In particular, CDH5 seems to synergically support Wnt/β-catenin signaling to enhance TCF activity [81] when added between days 5 and 7. However, the mechanism remains to be clarified because CDH5 may inhibit Wnt/β-catenin signaling sequestering β-catenin into cadherin complex at the cell surface and preventing TCF activation [82].

**Figure 4 ijms-25-03387-f004:**
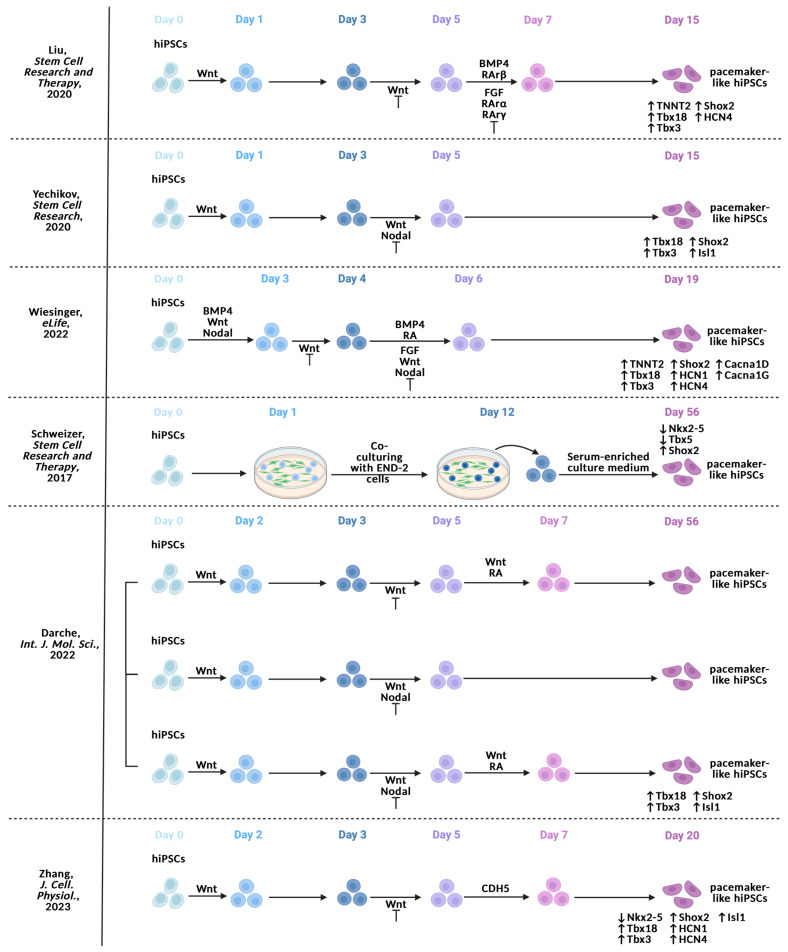
Summary of experimental designs to induce hiPSC-PMs through transgene-free methods [69,75,76,77,78,79].

Given that transgene-dependent or independent protocols, deciphering and combing the complex interplay between BMP4, RA, Nodal, and Wnt/β-catenin signaling pathways in SAN development holds the key to unable improved methods to obtain differentiated hiPSC-CM cultures containing high percentages of hiPSC-PMs [23,80,83].

## 5. Electrophysiological Properties of hiPSC-PMs

Besides cardiac development markers, ion channels confer electrophysiological properties that can be studied by the patch clamp technique to affirm the cellular identity expected from hiPSC-PMs. Native SAN has a unique characteristic of spontaneously beating rapidly. However, one of the main issues with hiPSC-CMs is that almost all of the cardiac subtypes (i.e., ventricular-, atrial- and hiPSC-PMs) have spontaneous activity [10,76,84]. The best way to analyze the functional characteristics of the supposed hiPSC-PMs is to deeply delineate the ion channels profile by combining information obtained from APs recordings (Table 1).

In general, hiPSC-PMs have faster AP rate and slower maximum upstroke velocity (dV/dt_max_) [65,67,68,69,75,76,77,79] compared to atrial- and ventricular-like hiPSC-CMs. As mentioned above, HCN1 and HCN4 contribute to the generation of I_f_ in the human SAN. Expression of I_f_ is considered as the electrophysiological hallmark to be tested in hiPSC-PM differentiation protocols [65,67,76,77,79]. I_f_ is normally recorded with regular voltage-clamp protocols [65,67,79], but sometimes I_f_ is assessed indirectly by treating cells with a selective I_f_ inhibitor (Ivabradine) and measuring the AP rate reduction [76,77]. However, the expression of I_f_ alone is not sufficient to define a pacemaker-like cell because automaticity relies also on the expression of voltage-gated L- and T-type Ca^2+^ channels (i.e., Ca_v_1.3 and Ca_v_3.1). To date, the functional expression of voltage-gated Ca^2+^ channels isoforms is poorly defined in hiPSC-PMs. Despite different groups showing the expression of different Ca^2+^ channel isoforms, patch clamp recordings of I_Ca_ are still focused on total Ca^2+^ current, without making a distinction between the two L-type isoforms Ca_v_1.2 and Ca_v_1.3 [67]. Notably, Ca_v_1.3 channels activate at more negative potentials compared to Ca_V_1.2 and are responsible for a large fraction of the total L-type Ca^2+^ current (I_CaL_) in the SAN [1]. In the native SAN, both Ca_v_1.3 and Ca_v_1.2 are expressed, whereas Ca_v_1.2 is the only isoform of the adult working myocardium. The fact that the expression of I_CaL_ activatesat negative voltages (−50, −40 mV), just as in the native SAN [2], is suggestive of a more pacemaker-like profile in hiPSC-PMs compared to ventricular- and atrial-like hiPSCs [67]. Recently, the black mamba toxin calciseptine [85] was found to be a selective inhibitor of Ca_v_1.2 vs. Ca_v_1.3 isoform and it could be a useful pharmacological tool to use when studying the relative expression of Ca_v_1.3 in hiPSC-CMs. Consistent with a SAN-like phenotype, hiPSC-PMs are characterized by a reduced I_Na_ current in comparison with ventricular-like hiPSCs [67]. This is in line with a reduction in the SAN dV/dt_max_, which is mainly dependent on Ca^2+^ channels [1,54]. In addition, the expression of I_KAch_ is also evaluated and increases in hiPSC-PMs compared to ventricular-like ones [67]. Finally, because adrenergic and muscarinic receptors are critical for the modulation of the response rate in the SAN [1], pharmacological regulation with Isoproterenol (β-AR stimulator) or M2R activators may be useful for assessing the proper functioning of positive–negative chronotropic machinery [67,77,78]. In summary, even if no single marker or electrophysiological hallmark has been reported as sharply defining the pacemaker-like differentiation fate of hiPSC-CMs, the available data suggest an association between action potential waveform, the steepness of diastolic depolarization, and expression of sinoatrial node ionic currents (i.e., I_KACh_, I_f_, I_CaL_, I_CaT_), providing a reliable criterion for defining this pathway. Further research will be required to define more discriminatory marker(s).

## 6. Conclusions

Nowadays, hiPSC-CMs are widely used as tool for disease modeling and represent valid platforms for the pharmacological screening of molecules and drugs. However, hiPSC-CMs have been mostly limited to model-inherited cardiopathies, which primarily affect ventricular cardiomyocytes. Hence, the current knowledge in the field mainly focuses on differentiating ventricular-like hiPSC-CMs. It is important to develop an in vitro multiscale model of the human SAN based on hiPSC-CM and hiPSC-PM differentiation protocols. From this perspective, the recent development of multi-chamber cardioids derived from hiPSC-CMs [86] may constitute an important step forward in producing 3D models in which pacemaker-like myocytes, derived, for example, from patients carrying sinoatrial node dysfunction, are fused with atrial and ventricular chamber-like tissue. This would be of paramount importance in terms of further understanding the mechanisms underlying sinus node dysfunction and testing innovative pharmacologic or molecular strategies translatable to clinics. In addition, the ability to generate hiPSC-PMs offers the potential to create biological pacemakers and specialized conduction tissues for treating patients with conduction system failures. Variability in the differentiation of hiPSC-CM subpopulations and the degree of cellular maturation may constitute a limitation. In this regard, the manipulation of cardiac development pathways appears a reliable method to drive hiPSCs towards an SAN phenotype capable of reducing variability. From this perspective, this review highlights the significant strides made in the past decade in generating hiPSC-PMs and emphasizes the critical contribution of insights gained from developmental biology to these achievements. The regulation of Wnt/β-catenin signaling, Nodal, RA, and BMP4 pathways seems to be fundamental to determining cardiac cells’ fates. However, many authors do not directly report data from differentiation time points to quantify the relative degree of efficiency of a given protocol to differentiate pacemaker-like cardiomyocytes in a mixed population of hiPSCs-CMs. Indeed, authors evaluate the efficiency of differentiation as the relative increase in expression of molecular markers of the sinoatrial node in hiPSCs-PMs compared to differentiated hiPSCs-CMs in parallel cultures, rendering comparison between different protocols difficult. Future work will thus require a coupled molecular and functional analysis of hiPSC-PM to ensure that they faithfully relate to the phenotypes of donor patients. This analysis will be essential in advancing the potential clinical applications of hiPSC-PMs and their relevance in modeling human cardiac conditions and diseases.

## Figures and Tables

**Figure 1 ijms-25-03387-f001:**
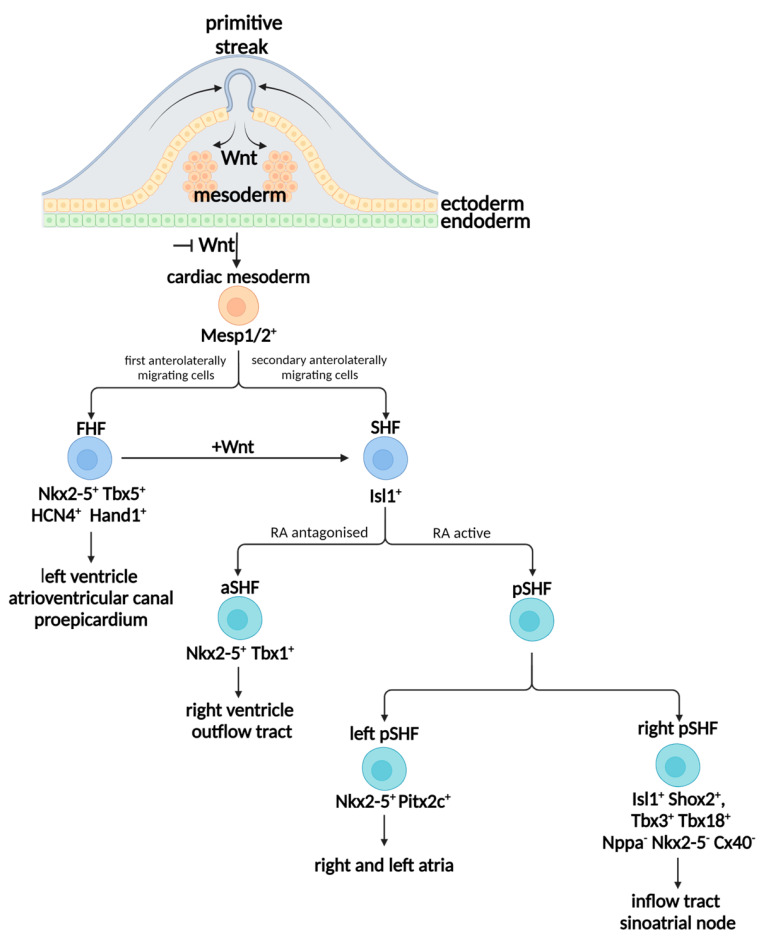
Generation of cardiac mesoderm during gastrulation and schematic representation of cardiovascular lineage diversification. FHF, first heart field; SHF, second heart field; RA, retinoic acid.

**Figure 2 ijms-25-03387-f002:**
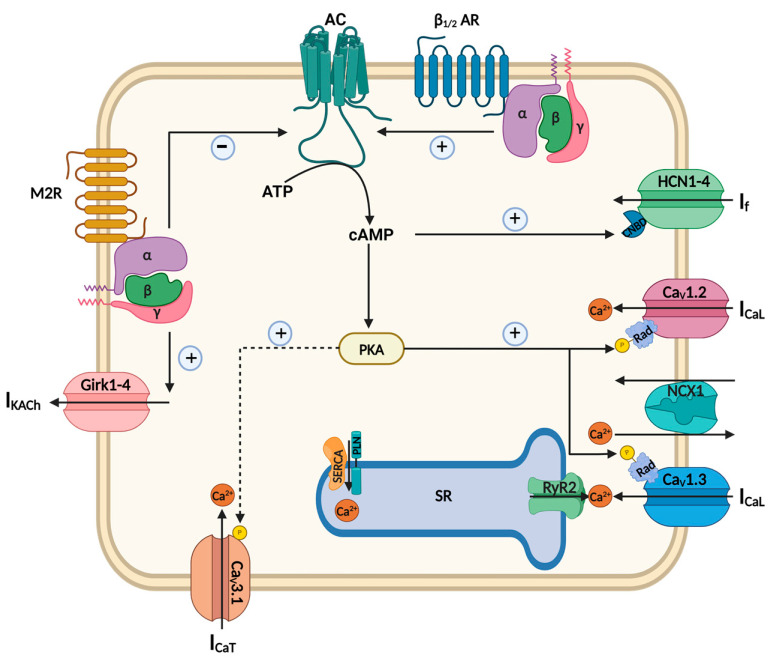
Main electrophysiological mechanisms of adult SAN cells are represented, including signaling pathways involved in adrenergic and muscarinic regulation of pacemaker activity. SERCA, sarco/endoplasmic reticulum Ca^2+^—ATPase; PLN, phospholamban.

**Table 1 ijms-25-03387-t001:** Summary of electrophysiological parameters evaluated in the different protocols.

	Birket, *Nat. Biotechnol.*, 2015 [65]	Protze, *Nat. Biotechnol.*, 2017 [67]	Schweizer, *Stem Cell Research and Therapy*, 2017 [77]	Ren, *Developmental Cell*, 2019 [68]	Liu, *Stem Cell Research and Therapy*, 2020 [69]	Yechikov, *Stem Cell Research*, 2020 [75]	Wiesinger, *eLife*, 2022 [76]	Zhang, *J. Cell. Physiol.*, 2023 [79]
**AP rate**	+	+	+	+	+	+	+	+
**dV/dt_max_**	+	+				+	+	+
**I_f_**	+	+	+ (indirectly)				+ (indirectly)	+
**I_CaL_**								
**I_CaT_**		+						
**I_Na_**		+						
**I_KACh_**		+						
**Adrenergic response**		+	+					
**Muscarinic response**		+	+					

## Data Availability

Not applicable.

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
