# Peer review of "State-of-the-Art Differentiation Protocols for Patient-Derived Cardiac Pacemaker Cells"

_ijms, 2024, doi:10.3390/ijms25063387_

Round 1

Reviewer 1 Report

Comments and Suggestions for Authors

The authors provide a comprehensive report with exceptionally clear figures on the use of human induced-pluripotent stem cell-derived cardiomyocytes and in particular, purported "pacemaker" subtypes. A few central challenges prohibit a complete understanding of this body of work:

1. The first sentence of the abstract is confusing and hinders understanding of the entire abstract. Abbreviations are not defined in the abstract.

2. The discussion of animal models for human cardiovascular disease is distracting. The field of hiPSCs is sufficiently established in this reviewer's estimation and requires no comparison to organism-models for justification. Perhaps the authors consider trimming/removing altogether?

3. Though the introduction concludes with a succinct description regarding the review's contents, it would be ideal for the article to objectively state what the uniqueness/novelty of the publication is, to assist in understanding the article's purpose.

4. One central concept that challenges this reviewer's understanding of the definition of hiPSC-PM is that non-hiPSC-PMs (meaning atrial and/or ventricular hiPSC-CMs) can exhibit spontaneous depolarizations (example: Chinyere IR, Bradley P, Uhlorn J, Eason J, Mohran S, Repetti GG, Daugherty S, Koevary JW, Goldman S, Lancaster JJ. Epicardially Placed Bioengineered Cardiomyocyte Xenograft in Immune-Competent Rat Model of Heart Failure. Stem Cells Int. 2021 Jul 24;2021:9935679. doi: 10.1155/2021/9935679. PMID: 34341667; PMCID: PMC8325579.)

Though the expression pattern certainly allows for lineage specification, it is not clear if a phenotypic/functional feature differentiates contractile from autorhythmic hiPSC-derived cardiac cells, that do not undergo mechanical maturation.

5. This reviewer is not a fan of including new (or any, for that matter) citations within the conclusion section as this final portion of the article should serve as a brief and succinct recap, rather than the explanation of new topics. Please consider revising.

Comments on the Quality of English Language

Although minor, some regions of the article contain awkward syntax. The authors would benefit from having a native English speaker carefully proofread the article to identify and improve some of the transitions within the abstract and the body of the article.

Author Response

The authors provide a comprehensive report with exceptionally clear figures on the use of human induced-pluripotent stem cell-derived cardiomyocytes and in particular, purported "pacemaker" subtypes. A few central challenges prohibit a complete understanding of this body of work:

  1. The first sentence of the abstract is confusing and hinders understanding of the entire abstract. Abbreviations are not defined in the abstract.

Thank you for your comment. We have now edited the abstract to better convey the overall message on the use of hiPSC derived cardiomyocytes. The first sentence is now removed (line 12-13-14 in red), and the abbreviations are defined.

  1. The discussion of animal models for human cardiovascular disease is distracting. The field of hiPSCs is sufficiently established in this reviewer's estimation and requires no comparison to organism-models for justification. Perhaps the authors consider trimming/removing altogether?

According to your suggestion, we have now trimmed the abstract (line 12-13-14 in red), as well as the conclusion (line 450-451-452 in red).  

  1. Though the introduction concludes with a succinct description regarding the review's contents, it would be ideal for the article to objectively state what the uniqueness/novelty of the publication is, to assist in understanding the article's purpose.

According to your suggestion, we have now rephrased the last paragraph of the Introduction section (see line 79 to 85 in yellow) to state and explain the scope and usefulness of our manuscript. Thank you for your suggestion.

  1. One central concept that challenges this reviewer's understanding of the definition of hiPSC-PM is that non-hiPSC-PMs (meaning atrial and/or ventricular hiPSC-CMs) can exhibit spontaneous depolarizations (example: Chinyere IR, Bradley P, Uhlorn J, Eason J, Mohran S, Repetti GG, Daugherty S, Koevary JW, Goldman S, Lancaster JJ. Epicardially Placed Bioengineered Cardiomyocyte Xenograft in Immune-Competent Rat Model of Heart Failure. Stem Cells Int. 2021 Jul 24; 2021:9935679. doi: 10.1155/2021/9935679. PMID: 34341667; PMCID: PMC8325579.)

Though the expression pattern certainly allows for lineage specification, it is not clear if a phenotypic/functional feature differentiates contractile from autorhythmic hiPSC-derived cardiac cells, that do not undergo mechanical maturation.

At present, no functional or electrophysiological marker has been widely accepted as the critical marker to distinguish contractile from autorhythmic hiPSC-derived cardiac cells, that do not undergo mechanical maturation. We believe that further research is required to identify such marker(s). In this manuscript we highlight that association between action potential waveform, steepness of diastolic depolarization and expression of sinoatrial ionic currents (IKACh, If, ICaL, ICaT…) provide a reliable criterion for defining this differentiation pathway. According to your suggestion, we now include a summary statement to clarify this point (from line 439 to 444 in yellow).

  1. This reviewer is not a fan of including new (or any, for that matter) citations within the conclusion section as this final portion of the article should serve as a brief and succinct recap, rather than the explanation of new topics. Please consider revising.

Citations removed from this section, with the exception of PMID 38029745 on cardioids, since it is adapted to the conclusion section. 

Comments on the Quality of English Language

Although minor, some regions of the article contain awkward syntax. The authors would benefit from having a native English speaker carefully proofread the article to identify and improve some of the transitions within the abstract and the body of the article.

Thank you for your comment. The manuscript underwent English proofreading by a native English speaker: Dr. Stefan Dubel. See Acknowledgments. 

Reviewer 2 Report

Comments and Suggestions for Authors
  1. The paper seems to focus primarily on summarizing available protocols without providing a critical evaluation or comparison of their effectiveness, efficiency, or limitations. Including such comparative analysis would enhance the paper's depth and utility for researchers.
  2. While the importance of generating pacemaker-like cells is highlighted, the paper could benefit from a more detailed discussion on the technical challenges associated with this process. Addressing these challenges would provide insights into the current limitations of the field and areas for future research.
  3. The paper does not discuss potential ethical considerations associated with hiPSC-derived research, such as informed consent, patient privacy, and the use of human tissues. Integrating ethical discussions into the paper would demonstrate a comprehensive understanding of the broader implications of the research.
  4. The focus appears to be solely on the technical aspects of obtaining pacemaker-like cells, overlooking broader implications such as clinical translation, scalability, and potential challenges in implementing these cells in therapeutic strategies. Expanding the scope to encompass these aspects would provide a more holistic view of the topic.
Comments on the Quality of English Language

no

Author Response

  1. The paper seems to focus primarily on summarizing available protocols without providing a critical evaluation or comparison of their effectiveness, efficiency, or limitations. Including such comparative analysis would enhance the paper's depth and utility for researchers.

Thank you for giving us the possibility of clarifying this point. Unfortunately, most authors do not directly report quantitative data at different time points to clearly define the relative degree of effectiveness/ efficiency of a given protocol to differentiate pacemaker-like cardiomyocytes in a mixed population of hiPSCs-CMs. Thus, in the first version of the manuscript, we decided not to discuss this point directly in the absence of usable data coming from the original publications. In original publications, “efficiency” is mostly reported as an absolute increase in expression of molecular markers of the sinoatrial node in hiPSCs-PM compared to differentiated hiPSCs-CM in the same cultures. Consequently, it is not possible to directly compare percentages of different protocols. We include a statement to point out this aspect (line 470 to 475 in yellow). 

  1. While the importance of generating pacemaker-like cells is highlighted, the paper could benefit from a more detailed discussion on the technical challenges associated with this process. Addressing these challenges would provide insights into the current limitations of the field and areas for future research.

In the manuscript (paragraph 4.1. and 4.2.) we discussed the need of introducing genetic modification to hiPSCs to maintain cellular population and select for differentiating hiPSC-PMs, while current methods solely based on small molecules complicates cellular selection. This conundrum has no clear-cut resolution in the present literature, so we could not deepen the discussion in. Nevertheless, in the conclusions section, we have highlighted some future technical challenges and avenues for improvement of the approach toward differentiation of hiPSC-PMs (line 454 to 468 in yellow). 

  1. The paper does not discuss potential ethical considerations associated with hiPSC-derived research, such as informed consent, patient privacy, and the use of human tissues. Integrating ethical discussions into the paper would demonstrate a comprehensive understanding of the broader implications of the research. Thank you for your suggestion.

We now summarize in the introduction section the main ethical issues related to collection and use of human cells to differentiate cardiomyocytes from hiPSCs. We have mentioned, in particular, ethical issues related to the benefit for the patient from basic research using hiPSC-CMs. New paragraph from line 54 to line 74.

  1. The focus appears to be solely on the technical aspects of obtaining pacemaker-like cells, overlooking broader implications such as clinical translation, scalability, and potential challenges in implementing these cells in therapeutic strategies. Expanding the scope to encompass these aspects would provide a more holistic view of the topic.

We have rewritten the conclusion section to implement the text with some considerations about the current and future translatability of this approach (see lines 454 to 460 in yellow), especially in relation to recent introduction of cardioid-based models. This approach may transfer conventional monolayer cultures of hiPSCs-PM to multi-scale level by incorporation into such cardioids. According to your indication, we identify as a major current challenge the reduction in variability of cellular differentiation and suggest concomitant analysis of both molecular and electrophysiological markers as a potential guideline to reduce variability and transfer the technology to multi-scale organoids (lines 454 to 478 in yellow).